# Geometry-Guided Diffusion Model with Masked Transformer for Robust Multi-View 3D Human Pose Estimation

Xinyi Zhang
Tsinghua Shenzhen International
Graduate School
Shenzhen, China
xinyi-zh22@mails.tsinghua.edu.cn

Qinpeng Cui
Tsinghua Shenzhen International
Graduate School
Shenzhen, China
cqp22@mails.tsinghua.edu.cn

Qiqi Bao
Zhejiang University of Science &
Technology
Hangzhou, China
nora919530829@163.com

Wenming Yang
Tsinghua Shenzhen International
Graduate School
Shenzhen, China
yang.wenming@sz.tsinghua.edu.cn

Qingmin Liao*
Tsinghua Shenzhen International
Graduate School
Shenzhen, China
liaoqm@tsinghua.edu.cn

## Abstract

Recent research on Diffusion Models and Transformers has brought significant advancements to 3D Human Pose Estimation (HPE). Nonetheless, existing methods often fail to concurrently address the issues of accuracy and generalization. In this paper, we propose a **G**eometry-guided Di**ff**usion Model with Masked Trans**former** (Masked Gifformer) for robust multi-view 3D HPE. Within the framework of the diffusion model, a hierarchical multi-view transformer-based denoiser is exploited to fit the 3D pose distribution by systematically integrating joint and view information. To address the long-standing problem of poor generalization, we introduce a fully random mask mechanism without any additional learnable modules or parameters. Furthermore, we incorporate geometric guidance into the diffusion model to enhance the accuracy of the model. This is achieved by optimizing the sampling process to minimize reprojection errors through modeling a conditional guidance distribution. Extensive experiments on two benchmarks demonstrate that Masked Gifformer effectively achieves a trade-off between accuracy and generalization. Specifically, our method outperforms other probabilistic methods by > 40% and achieves comparable results with state-of-the-art deterministic methods. In addition, our method exhibits robustness to varying camera numbers, spatial arrangements, and datasets.

## CCS Concepts

• **Computing methodologies** → *Activity recognition and understanding*.

## Keywords

Multi-view 3D human pose estimation, diffusion models, transformers

---

*Corresponding author

ACM Reference Format:
Xinyi Zhang, Qinpeng Cui, Qiqi Bao, Wenming Yang, and Qingmin Liao. 2024. Geometry-Guided Diffusion Model with Masked Transformer for Robust Multi-View 3D Human Pose Estimation. In *Proceedings of the 32nd ACM International Conference on Multimedia (MM '24), October 28-November 1, 2024, Melbourne, VIC, Australia.* ACM, New York, NY, USA, 10 pages. https://doi.org/10.1145/3664647.3681265

## 1 Introduction

3D Human Pose Estimation (HPE) aims to accurately localize sparse joints on the human skeleton in 3D space through analysing the multimedia data such as images and videos. The obtained geometric and positional information of the human body can be beneficial for various downstream tasks [57], including action recognition [40, 51, 59], action prediction [3, 44, 54], action correction and online coaching [8, 9, 42], animation [10, 25, 47], and augmented reality/virtual reality [46, 50, 52].

Depending on the number of views, 3D HPE can be categorized into single-view and multi-view methods. Single-view methods [23, 24, 33, 53, 56] estimate 3D human pose from monocular images or videos, which may suffer from depth ambiguity. Such challenge can be alleviated to some extent when using multiple synchronized and calibrated cameras. Therefore, more attention has been paid to multi-view methods. Multi-view 3D HPE methods primarily follow a two-stage paradigm: (1) Utilizing off-the-shelf 2D pose detectors [4, 49] to locate 2D positions of joints in each view; (2) Mapping 2D positions to 3D poses and fusing multi-view information selectively. In this work, we focus on the latter step, known as the 2D-to-3D lifting process, following recent processes [18, 29, 34, 58].

2D-to-3D lifting methods accurately predict 3D poses from multi-view 2D poses by designing effective neural networks. Many approaches [13, 27, 28, 34] leverage the powerful long-range dependency and interaction modeling capabilities of Transformers [41] to fuse multi-view geometric information, thereby minimizing reprojection errors. However, these methods still face several challenges. Firstly, strictly constrained by camera numbers and spatial arrangements, **it is difficult to generalize well to an arbitrary number of unseen views.** Moreover, these methods are generally *deterministic*, inferring the most probable single pose directly from the given input, which may fall into local optima in complex scenarios.

*Probabilistic methods* [1] can alleviate uncertainty by generating multiple plausible pose hypotheses, leading to more accurate pose estimation. They have been extensively studied in single-view 3D HPE [2, 11, 15, 24, 33], which is an ill-posed problem due to depth ambiguity brought on by single-view data. However, uncertainties persist in multi-view scenarios from severe occlusions, self-occlusions and noisy 2D detections. Therefore, probabilistic models for multi-view are worthy of further investigation. Existing methods directly utilize Diffusion Models (DMs) [14, 36, 37] to generate multiple hypotheses. **Due to the lack of geometric guidance, the high-accuracy potential remains underexplored.**

To address the above issues, we propose a **G**eometry-guided D**if**fusion Model with Masked Trans**former** (Masked Gifformer), which unifies the geometry-guided conditional diffusion generative process with the masked hierarchical multi-view transformer, facilitating accurate distribution forecasting for 3D poses while enhancing generalization capabilities. As illustrated in Figure 1, we use the framework of DDPM [14]. Initially, noise of varying scales is added to the ground truth 3D poses. Subsequently, a transformer-based denoiser is exploited to approximate the probability distribution $p(X^{3D} \mid X^{2D})$ of 3D coordinates $X^{3D}$ given the detected 2D coordinates $X^{2D}$. During the inference, Masked Gifformer employs its geometry-guided diffusion process to iteratively refine pose predictions.

Motivated by [53], we use a Hierarchical Multi-view Fusion method which aggregates joint and view information separately. More specifically, a Joint Transformer Block (JTB) is employed to learn physical constraints among joints within each view. Then, a Multi-view Transformer Block (MTB) is used to allow each joint to independently learn the corresponding information from other views. These two transformer blocks are trained alternately, facilitating comprehensive multi-view information fusion. **To address the problem of poor generalization, we introduce a fully random mask mechanism into MTB without any additional learnable modules or parameters.** By randomly masking a portion of views during training, it is possible to avoid the learned features that are highly correlated with camera numbers and positions. Different from [34], our model even discards information from the current view. This strategy promotes a more effective integration of features from alternative views instead of overfitting to features from the source dataset, thus exhibiting robustness across different datasets. The efficiency of self-attention computations in Transformers can also be improved due to introducing sparsity.

During the inference, we sample multiple noise instances from a Gaussian distribution as initial 3D pose hypotheses. These pose hypotheses are then fed into the transformer-based denoiser conditioned on multi-view 2D keypoints. To further enhance the accuracy, we incorporate geometric knowledge into the sampling process. Inspired by [7, 43], we model the $X^{3D}$-conditioned guidance distribution $p(X^{2D} \mid X^{3D})$ as a negative exponential form of reprojection errors and use its gradient $\nabla_{X^{3D}} \log p\left(X^{2D} \mid X^{3D}\right)$ to perturb the pre-trained predictions. **This strategy guides the sampling process towards minimizing reprojection errors, thereby further enhancing the accuracy of pose estimation.** Finally, a single accurate and robust 3D pose is generated by aggregating all of hypotheses for practice use.

In summary, our main contributions are as follows:

- We propose a **G**eometry-guided D**if**fusion Model with Masked Trans**former** (Masked Gifformer) for robust multi-view 3D HPE. This method combines the capabilities of Transformers for high-quality multi-view information fusion and the probabilistic nature of Diffusion Models for generating multiple robust 3D pose hypotheses.
- We introduce a fully random mask mechanism into the multi-view transformer block, enhancing its strong robustness to camera numbers, spatial arrangements and datasets without adding any learnable modules or parameters.
- We incorporate geometric information into the sampling process of diffusion models, guiding it towards minimizing reprojection errors.
- Extensive experiments on two popular 3D HPE benchmarks demonstrate the effectiveness of Masked Gifformer. It not only achieves state-of-the-art results in precision but also exhibits strong generalization capabilities across different camera views, spatial placements, and datasets.

## 2 Related Works

**Transformer-based 3D HPE.** With the superior capability of Transformers [41] in modeling long-range dependencies, numerous Transformer-based single-view [22, 23, 53] and multi-view [13, 27, 28, 34] 3D HPE methods have emerged. In multi-view settings, researchers have employed epipolar geometry for information fusion [13, 27] to minimize reprojection errors. Specifically, He et al. [13] extracted multiple features along the epipolar line in the source view and then fused them with features from the reference view. To fully exploit the semantic information in the source view, Ma et al. [27] proposed the concept of epipolar field, merging more information near the epipolar line while maintaining epipolar constraints. However, these methods cannot generalize well to an arbitrary number of unseen views as constrained by camera numbers and spatial arrangements. To address this issue, Shuai et al. [34] introduced a random mask mechanism within Transformers, enabling them to adaptively handle varying numbers of views and video lengths. Nevertheless, this mechanism does not guarantee generalization to different datasets. Jiang et al. [18] improved cross-dataset generalization by modelling camera distributions. However, this method employed 2D heatmaps as feature inputs and used computationally intensive voxel networks during pose reconstruction, making it impractical for real-time applications.

**Diffusion-based 3D HPE.** The aforementioned Transformer-based methods often infer the most probable single pose from the given input, falling under *deterministic methods*. They may be trapped in local optima, especially in complex scenarios. *Probabilistic methods* achieve more accurate pose estimation by generating multiple plausible pose hypotheses. Due to the inherent probabilistic nature of DMs, numerous diffusion-based single-view 3D HPE methods have emerged [2, 11, 15, 24, 33] to mitigate the uncertainty brought by depth ambiguity. Shan et al. [33] directly applied DMs to single-view 3D HPE. Cai et al. [2] introduced a disentanglement strategy in the forward process of the diffusion model to integrate the explicit human body prior. However, high-precision potential

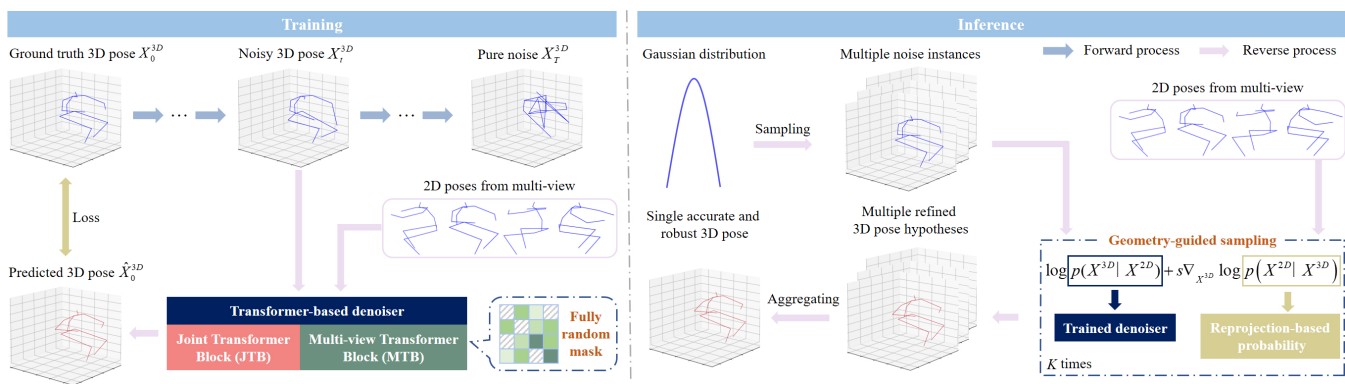

**Figure 1: Overview of Masked Gifformer. During training, a transformer-based denoiser is supervised to learn $p(X^{3D} \mid X^{2D})$ within the framework of diffusion models. JTB and MTB are trained alternately to fuse joint and viewpoint information hierarchically. With the proposed fully random mask mechanism embedded within MTB, our model can generalize to varying camera numbers, spatial arrangements and datasets. During the inference, by modeling the conditional guidance distribution $p(X^{2D} \mid X^{3D})$, multiple 3D pose hypotheses are sampled towards minimizing reprojection errors.**

needs further exploration as these methods lack the guidance of geometric information. In multi-view scenarios, 3D HPE still faces uncertainties arising from severe occlusions or self-occlusions, noisy 2D detections, etc. Bartol et al. [1] proposed a multi-view triangulation stochastic framework, generating multiple pose hypotheses to overcome occlusions. Research on diffusion-based multi-view 3D HPE is limited, which warrants further investigation.

**Summary.** Both Transformers and DMs demonstrate significant potential in 3D HPE. To benefit from high-quality multi-view information fusion and multiple high-precision 3D pose hypotheses generation, we explore an approach that integrates Transformers and DMs. We introduce a fully random mask mechanism to enhance the generalization capabilities across varying camera numbers, spatial arrangements, and cross-datasets in a lightweight manner. Additionally, we incorporate geometric information into the sampling process of the diffusion model to guide it towards minimizing reprojection errors.

## 3 Method

We consider the problem of estimating 3D positions of the human body when multi-view cameras are precisely calibrated.

Formally, given 2D joints $X^{2D} = \left\{ x_i^{2D} \in \mathbb{R}^{J \times 2} \mid i = 1 \dots N \right\}$, our goal is to predict 3D positions $X^{3D} = \left\{ x_i^{3D} \in \mathbb{R}^{J \times 3} \mid i = 1 \dots N \right\}$ in the world coordinate system. $N$ and $J$ represent the number of cameras and joints, respectively. When cameras are calibrated with precisely known intrinsic and extrinsic parameters $Y$, deterministic methods leverage Transformers [13, 27, 28] denoted as $F$ to fuse multi-view geometric information, resulting in $X^{3D} = F(X^{2D}, Y)$.

However, due to occlusions or self-occlusions, some joints may not be visible in the given multiple views, resulting in non-uniqueness of the human pose. The noisy 2D joints further exacerbates the uncertainty of HPE. Therefore, it is necessary to use *probabilistic methods* to generate multiple ($H$) hypotheses of human poses $\{X_i^{3D}\}_{i=1}^{H}$ and find the most probable one.

Combining the probabilistic nature of DMs with multi-view fusion capabilities of Transformers, we propose Masked Gifformer

for robust multi-view 3D HPE. As shown in Figure 1, we train a transformer-based denoiser to fit the probability distribution $p(X^{3D} \mid X^{2D})$ within the framework of diffusion models. With the proposed fully random mask strategy, our model is robust to varying camera numbers, spatial arrangements and datasets. During the inference, by modeling the conditional guidance distribution $p(X^{2D} \mid X^{3D})$, multiple 3D pose hypotheses are sampled towards minimizing reprojection errors. Finally, a single accurate and robust 3D pose is obtained by aggregating all of hypotheses.

### 3.1 Preliminaries of Diffusion Models

DMs [6, 14, 36, 38, 39] have been demonstrated to possess promising capabilities in modelling complex data distributions. A Denoising Diffusion Probabilistic Model (DDPM) [14] is one of them, which encompasses forward and reverse processes.

Specifically, in the forward process, DDPM incrementally introduces Gaussian noise within $T$ steps to perturb the input $\{x_t\}_{t=0}^{T}$. Such noising process forms a Markov chain and satisfies:

$$q\left(x_t \mid x_{t-1}\right) = \mathcal{N}\left(x_t; \sqrt{1 - \beta_t} x_{t-1}, \beta_t \mathbb{I}\right). \tag{1}$$

Here, $\{\beta_t\}_{t=1}^{T} \in (0, 1)$ is a monotonically increasing set of noise scales with diffusion steps $T$. $\mathbb{I}$ represents the identity matrix, and $\mathcal{N}(x; \mu, \sigma)$ denotes the Gaussian distribution generating $x$ with mean $\mu$ and covariance $\sigma$. Let $\alpha_t = 1 - \beta_t$ and $\bar{\alpha}_t = \prod_{i=1}^{t} \alpha_i$, then noisy samples $x_t$ can be obtained in a single step:

$$x_t \sim q\left(x_t \mid x_0\right) = \mathcal{N}\left(x_t; \sqrt{\bar{\alpha}_t} x_0, (1 - \bar{\alpha}_t) \mathbb{I}\right). \tag{2}$$

If $\bar{\alpha}_t \rightarrow 0$, the distribution of $x_T$ should closely approximate the standard Gaussian distribution, i.e., $q(x_T) = \mathcal{N}(0, \mathbb{I})$.

In the reverse process, DDPM learns the mapping from a simple normal distribution to the true data distribution. If $\{\beta_t\}_{t=1}^{T}$ is sufficiently small, $p_\theta\left(x_{t-1} \mid x_t\right)$ still follows a Gaussian distribution. Fixing the covariance as a constant, a neural network $\mathcal{D}_\theta(x_t, t)$ is trained to predict the mean of the Gaussian distribution, that is:

$$p_\theta\left(x_{t-1} \mid x_t\right) = \mathcal{N}\left(x_{t-1}; \sqrt{\alpha_t} \mathcal{D}_\theta\left(x_t, t\right), (1 - \alpha_t) \mathbb{I}\right), \tag{3}$$

where $\theta$ represents learnable parameters. Starting from the random pose $x_T \sim \mathcal{N}(\mathbf{0}, \mathbb{I})$, $\hat{x}_0$ conforming to the true distribution can be generated through iterative sampling:

$$x_{t-1} \sim \mathcal{N}\left(x_{t-1}; \sqrt{\bar{\alpha}_{t-1}} \mathcal{D}_\theta\left(x_t, t\right), (1 - \bar{\alpha}_{t-1})\mathbb{I}\right). \qquad (4)$$

## 3.2 Diffusion-Based Multi-View 3D HPE

In this section, we utilize DMs to model the probability distribution of 3D human poses across multiple views, and generate multiple pose hypotheses leveraging its probabilistic nature. Based on the diffusion framework [14] discussed in Sec.3.1, we first add noise to the ground truth 3D poses, and then train a denoiser $\mathcal{D}_\theta$ on a training set to fit the true data distribution. Note that given the 2D joint coordinates $X^{2D}$, we model the conditional probability distribution $p_\theta(X^{3D}|X^{2D})$. Therefore, unlike the forward process (Eq.(1)) which is independent of $X^{2D}$, the reverse process depends on the input 2D joint coordinates $X^{2D}$, i.e.,

$$p_\theta\left(X_{t-1}^{3D}|X_t^{3D}, X^{2D}\right) = \mathcal{N}\left(X_{t-1}^{3D}; \sqrt{\alpha_t}\mathcal{D}_\theta\left(X_t^{3D}, t, X^{2D}\right), (1-\alpha_t)\mathbb{I}\right). \quad (5)$$

**Denoiser $\mathcal{D}_\theta$.** Due to the promising information interaction and global aggregation capabilities of Transformers [41], we implement the denoiser $\mathcal{D}_\theta$ using Transformers. As illustrated in Figure 2, previous work [13, 27, 28, 34, 55] proposed hybrid multi-view fusion methods, using a single type of transformer block to integrate joint information across all views. Inspired by [53], we employ a hierarchical multi-view fusion method, using two distinct types of transformer blocks to focus on physical constraints among joints and the correlation across different views. Specifically, we first use a Joint Transformer Block (JTB) to learn the physical correlations between joints within each view. Subsequently, a Multi-view Transformer Block (MTB) is employed to enable each joint to individually learn corresponding information from other views. These two types of transformer blocks are alternately trained, achieving comprehensive multi-view information fusion.

**Fully random mask mechanism.** However, the aforementioned naive architecture imposes strict limitations on camera numbers and spatial arrangements, making it challenging to generalize well to an arbitrary number of unseen views. To address this issue, [34] proposed a random mask mechanism, which randomly masked the attention matrix except for the elements on the diagonal at the rate of $M$. However, experimental results in Table 6 demonstrate that such random mask mechanism exhibits poor generalization across datasets. Therefore, we propose a fully random mask mechanism as depicted in Figure 3(b), which even randomly discards information from the current viewpoint. This fully random mask mechanism offers the following advantages:

- By randomly masking a portion of views during training, it is possible to avoid the learned features that are highly correlated with camera numbers and positions. Therefore, our model demonstrates robust generalization to varying camera numbers and spatial arrangements.
- By discarding information from the current view, a more effective integration of features from alternative views is learned instead of overfitting to features from the source dataset, thus enhancing robustness across different datasets.
- By introducing sparsity, it can improve the efficiency of the self-attention computations in Transformers.

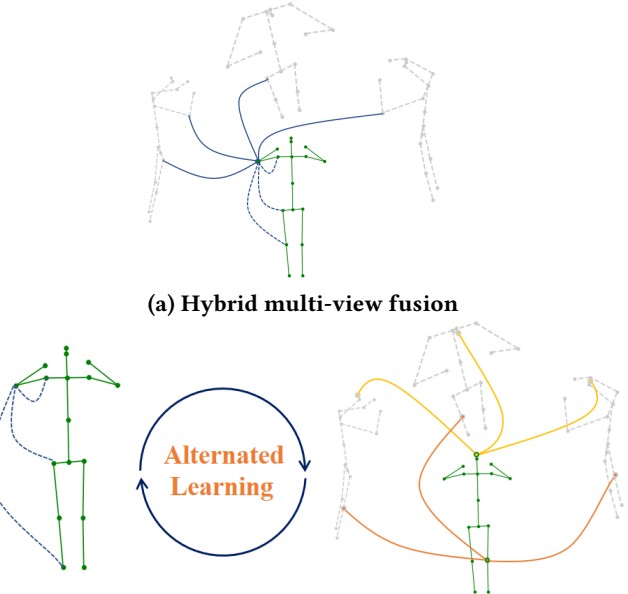

**(a) Hybrid multi-view fusion**

**(b) Hierarchical multi-view fusion**

**Figure 2: Different multi-view fusion methods. (a) Hybrid multi-view fusion [13, 27, 28, 34, 55]: fusing all joint information across all views by a single type of transformers. (b) Hierarchical multi-view fusion (Ours): fusing joint and view information separately by two distinct types of transformers.**

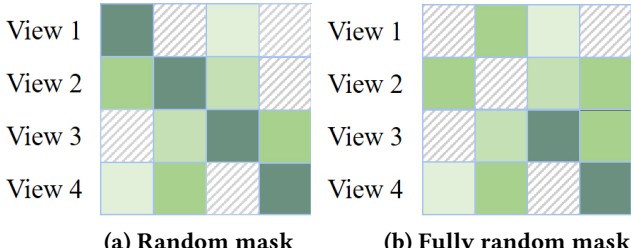

**(a) Random mask**     **(b) Fully random mask**

**Figure 3: Different random mask mechanisms. (a) Random mask mechanism [34]: randomly masking the attention matrix except for the elements on the diagonal. (b) Fully random mask mechanism (Ours): randomly masking the attention matrix including the elements from the same view. This mechanism ensures our model generalizes well to varying camera numbers, spatial arrangements, and datasets.**

Incorporating the proposed fully random mask mechanism into the MTB, the detailed structure of the denoiser is showed in Figure 4. The Masked Multi-head Self-attention (MMSA) is calculated by:

$$\text{MMSA}(\mathbf{X}) = \text{Softmax}\left(\frac{\mathbf{Q}\mathbf{K}^T \times \mathbf{M}}{\sqrt{D}}\right)\mathbf{V}, \qquad (6)$$

where the query $\mathbf{Q}$, the key $\mathbf{K}$, and the value $\mathbf{V}$ are obtained from the linear projections of the input $\mathbf{X}$. $\mathbf{M}$ is a fully random mask. $D$ is the dimension of feature.

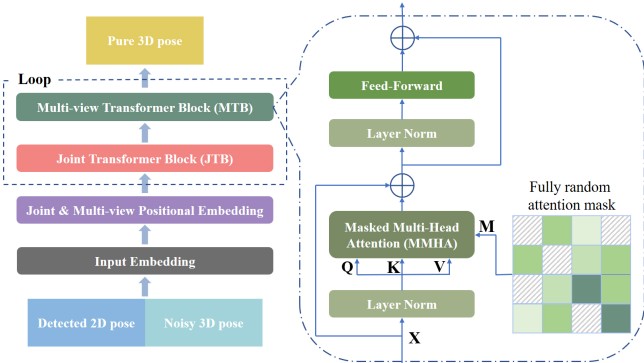

**Figure 4: Detailed structure of the transformer-based denoiser $\mathcal{D}_\theta$. JTB learns physical constraints between joints within each view. MTB enables each joint to individually learn corresponding information from other views. JTB and MTB are stacked for 16 loops and alternately trained, achieving comprehensive multi-view information fusion.**

**Training loss.** During training, denoiser $\mathcal{D}_\theta$ is supervised by denoising loss:

$$\mathcal{L} = E_{t\sim[1,T], X_t^{3D}\sim q(X_t^{3D}|X_0^{3D}, X^{2D})} \left\| \mathcal{D}\left(X_t^{3D}, t, X^{2D}\right) - X_0^{3D}\right\|_2^2, \quad (7)$$

where the expectation is taken over all diffusion time $t$, corresponding to diffused samples $X_t^{3D} \sim q\left(X_t^{3D} \mid X_0^{3D}, X^{2D}\right)$.

**Multi-Hypothesis Generation and Aggregation.** During the inference, given a new set of 2D joints coordinates from multi-view, Eq.(4) is rewritten as:

$$X_{t-1}^{3D} \sim \mathcal{N}\left(X_{t-1}^{3D}; \sqrt{\bar{\alpha}_{t-1}}\mathcal{D}_\theta\left(X_t^{3D}, t, X^{2D}\right), (1-\bar{\alpha}_{t-1})\,\mathbb{I}\right). \quad (8)$$

Starting from $H$ random poses sampled from the Gaussian distribution, samples $\{\hat{X}_{0i}^{3D}\}_{i=1}^H$ conforming to the true distribution can be generated by the iterative sampling shown in Eq.(8). Following [33], the reverse process iterates $K$ times to enhance the differences between hypotheses.

Since a single 3D pose is still required for practical use, we use the joint-wise reprojection-based multi-hypothesis aggregation method [33], which selects the optimal hypothesis at the joint level based on reprojection errors (3D → 2D).

## 3.3 Geometry-Guided Sampling

As discussed in Sec.3.2, DMs can directly map 2D poses to the 3D poses via the supervised denoiser. However, deep networks often exhibit poor performance in precise quantity regression and are prone to overfitting [21, 43]. Inspired by [43], we incorporate geometric information into the sampling process, significantly enhancing the accuracy of Masked Gifformer while maintaining its generalization capability. Focusing on the consistency between 2D screen coordinates and 3D world coordinates, we use reprojection errors to guide DDPM sampling iterations (Eq.(8)).

**Reprojection errors.** Given the 2D pose $X^{2D}$ and the predicted 3D pose $X^{3D}$, we compute errors between the projection of the predicted 3D world coordinates onto the 2D image plane and their true projection [12], i.e.,

$$e_{\text{reproj}} = \sum_{N}^{N} \sum_{J}^{J} ||X^{2D} - \frac{1}{\lambda}K \cdot T \cdot X^{3D}||_2^2. \quad (9)$$

$K = \left\{k_i \in \mathbb{R}^{3\times3} \mid i = 1\ldots N\right\}$ represents camera intrinsic matrices. $T = \left\{T_i = \begin{bmatrix} R_i & t_i \\ 0^T & 1 \end{bmatrix} \mid i = 1\ldots N\right\}$. $R_i \in \mathbb{SO}(3)$ and $t_i \in \mathbb{R}^3$ are the rotation matrix and translation vector, respectively. $K\cdot T$ projects 3D points $X^{3D}$ from world coordinates to 2D points $X^{2D}$ in the screen coordinates with $K\cdot T\cdot X^{3D} \sim \lambda\left[X^{2D}; 1\right]$, where $\sim$ indicates homogeneous equivalence. A smaller reprojection error indicates that the predicted 3D pose is more consistent geometrically with the 2D pose, enhancing the accuracy and reliability of predictions.

**Reprojection-guided sampling.** Following the classifier diffusion guidance [7], we guide the sampling towards minimizing reprojection errors, thereby satisfying the geometric consistency constraint from 3D to 2D.

Specifically, we model $X^{3D}$-conditioned guidance distribution $p(X^{2D} \mid X^{3D})$ as a negative exponential form of reprojection errors:

$$p\left(X^{2D} \mid X^{3D}\right) \propto \exp\left(-e_{\text{reproj}}\right). \quad (10)$$

During each sampling iteration, classifier-guidance perturbs the predicted mean with a gradient of the conditioned guidance distribution $p(X^{2D} \mid X^{3D})$, i.e.,

$$\hat{\mathcal{D}}_\theta\left(X^{3D}, t, X^{2D}\right) = \mathcal{D}_\theta\left(X^{3D}, t, X^{2D}\right) + s\nabla_{X^{3D}} \log p\left(X^{2D} \mid X^{3D}\right) \quad (11)$$

where $s$ controls the strength of the guidance. Finally, the perturbed mean $\hat{\mathcal{D}}_\theta\left(X^{3D}, t, X^{2D}\right)$ replaces $\mathcal{D}_\theta\left(X^{3D}, t, X^{2D}\right)$ in Eq.(5) and Eq.(8), thereby achieving the reprojection-guided sampling.

## 4 Experiments

### 4.1 Datasets and Metrics

Experiments are conducted on widely used 3D HPE benchmark datasets, including Human 3.6M [16] and CMU Panoptic [19, 20, 35].

**Human3.6M** is one of the largest motion capture datasets. It consists of videos captured from 11 professional actors and actresses performing 15 distinct activities (e.g., walking), with 4 calibrated and synchronized high-speed motion cameras placed at the corners of a rectangular room. Human3.6M dataset not only provides calibrated camera intrinsic and extrinsic parameters but also accurate 3D joint positions. Following [18, 33, 34], our model is trained on 5 subjects (S1, S5, S6, S7, and S8) and tested on S9 and S11.

**CMU Panoptic** stands out as a large-scale dataset that utilizes a larger number of cameras. It consists of 480 VGA cameras and 31 HD cameras installed on a hemispherical structure, capturing various activities performed by single or multiple individuals, such as dancing. Given our focus on single-person HPE, we only use HD videos from individual subjects, following [1, 48]. In accordance with [1, 18], we partition the dataset into four subsets (CMU1 ∼ CMU4), with each subset varying in camera numbers and arrangements. CMU Panoptic also provides calibrated camera parameters and 3D joint coordinates.

**Metrics.** Mean Per Joint Position Error (MPJPE) is used to evaluate the accuracy of 3D HPE. A lower MPJPE indicates higher accuracy in predicted joint positions relative to the ground truth.

**Table 1: Evaluation results on Human3.6M for the state-of-the-art deterministic and probabilistic methods. No additional training data setup. $\star$: using temporal information. $H$, $K$: the number of hypotheses and iterations. GS: geometric-guided sampling proposed by us. Best in bold.**

| Methods | Dir. | Disc. | Eat | Greet | Phone | Photo | Pose | Purch. | Sit | SitD. | Smoke | Wait | WalkD. | Walk | WalkT. | Mean |
|---|---|---|---|---|---|---|---|---|---|---|---|---|---|---|---|---|
| | | | | | | Deterministic Methods (Multi-view) | | | | | | | | | | |
| Qiu et al. [31] | 28.9 | 32.5 | 26.6 | 28.1 | 28.3 | 29.3 | 28.0 | 36.8 | 42.0 | 30.5 | 35.6 | 30.0 | 28.33 | 30.0 | 30.5 | 31.2 |
| Remelli et al.[32] | 27.3 | 32.1 | 25.0 | 26.5 | 29.3 | 35.4 | 28.8 | 31.6 | 36.4 | 31.7 | 31.2 | 29.9 | 26.9 | 33.7 | 30.4 | 30.2 |
| Jiang et al. [18] | 24.0 | 25.4 | 26.6 | 30.4 | 32.1 | **20.1** | 20.5 | 36.5 | 40.1 | 29.5 | 27.4 | 27.6 | **20.8** | 24.1 | **22.0** | 27.8 |
| He et al. [13] | 25.7 | 27.7 | 23.7 | 24.8 | 26.9 | 31.4 | 24.9 | 26.5 | 28.8 | 31.7 | 28.2 | 26.4 | 23.6 | 28.3 | 23.5 | 26.9 |
| Zhou et al. [58] ($\star$) | 24.8 | 27.7 | 24.3 | 24.9 | 27.7 | 29.8 | 24.5 | **25.3** | 30.5 | 33.4 | 28.2 | 24.0 | 28.4 | 24.7 | 24.3 | 26.5 |
| Shuai et al. [34] | 23.8 | 26.0 | 23.9 | 25.0 | 28.2 | 29.7 | 23.6 | 25.5 | 30.1 | 37.3 | 26.6 | 24.5 | 27.4 | 23.1 | 23.4 | 26.5 |
| Ma et al. [27] | 24.4 | 26.4 | 23.4 | **21.1** | 25.2 | 23.2 | 24.7 | 33.8 | 29.8 | 26.4 | 26.8 | 24.2 | 23.2 | 26.1 | 23.3 | 25.8 |
| Ma et al. [28] | **21.8** | 26.5 | **21.0** | 22.4 | **23.7** | 23.1 | 23.2 | 27.9 | 30.7 | **24.6** | 26.7 | 23.3 | 21.2 | 25.3 | 22.6 | **24.4** |
| Ours ($H = 1$, $K = 1$) | 22.5 | **24.3** | 23.8 | 23.1 | 26.7 | 25.1 | **22.0** | 25.5 | **28.3** | 30.3 | **24.8** | 23.1 | 26.7 | **22.4** | 22.6 | 24.7 |
| | | | | | | Probabilistic Methods (Single-view) | | | | | | | | | | |
| Wehrbein et al. [45] ($H = 200$) | 38.5 | 42.5 | 39.9 | 41.7 | 46.5 | 51.6 | 39.9 | 40.8 | 49.5 | 56.8 | 45.3 | 46.4 | 46.8 | 37.8 | 40.4 | 44.3 |
| Li et al. [23] ($H = 3$, $\star$) | 39.2 | 43.1 | 40.1 | 40.9 | 44.9 | 51.2 | 40.6 | 41.3 | 53.5 | 60.3 | 43.7 | 41.1 | 43.8 | 29.8 | 30.6 | 43.0 |
| Cai et al. [2] ($H = 20$, $K = 10$, $\star$) | 36.4 | 39.5 | 34.9 | 37.6 | 40.1 | 45.9 | 37.8 | 37.8 | 51.5 | 52.2 | 40.8 | 38.3 | 38.3 | 27.0 | 27.0 | 39.0 |
| Gong et al. [11] ($H = 5$, $K = 50$, $\star$) | 33.2 | 36.6 | 33.0 | 35.6 | 37.6 | 45.1 | 35.7 | 35.5 | 46.4 | 49.9 | 37.3 | 35.6 | 36.5 | 24.4 | 24.1 | 36.9 |
| Ci et al. [5] ($H = 200$) | **31.7** | 35.4 | 31.7 | 32.3 | 36.4 | 42.4 | **32.7** | 31.5 | **41.2** | 52.7 | 36.5 | 34.0 | 36.2 | 29.5 | 30.2 | 35.6 |
| Shan et al. [33] ($H = 20$, $K = 10$, $\star$) | 33.0 | 34.8 | 31.7 | 33.1 | 37.5 | 43.7 | 34.8 | 33.6 | 45.7 | 47.8 | 37.0 | 35.0 | 35.0 | 24.3 | 24.1 | 35.4 |
| [33]+GS ($H = 20$, $K = 10$, $\star$) | 31.8 | **33.2** | **30.0** | **31.9** | **34.5** | **40.9** | 34.0 | **31.8** | 43.1 | **46.5** | **35.1** | **33.2** | **33.3** | **23.0** | **23.0** | **33.7** |
| | | | | | | Probabilistic Methods (Multi-view) | | | | | | | | | | |
| Bartol et al. [1] ($H = 200$, $K = 500$) | 27.5 | 28.4 | 29.3 | 27.5 | 30.1 | 28.1 | 27.9 | 30.8 | 32.9 | 32.5 | 30.8 | 29.4 | 28.5 | 30.5 | 30.1 | 29.1 |
| Ours ($H = 20$, $K = 10$) | **15.7** | **17.4** | **17.5** | **16.0** | **18.9** | **18.9** | **15.2** | **16.2** | **21.5** | **22.0** | **17.8** | **15.6** | **18.0** | **14.7** | **16.0** | **17.4** |

**Table 2: Evaluation results on CMU Panoptic (CMU3 with four cameras). Best in bold.**

| Methods | MPJPE, mm |
|---|---|
| Deterministic Methods | |
| Jiang et al. [18] | 24.2 |
| Iskakov et al. Algebraic [17] | 21.3 |
| Moliner et al. [29] | 17.2 |
| Iskakov et al. Volumetric [17] | **13.7** |
| Ours ($H = 1$, $K = 1$) | 16.8 |
| Probabilistic Methods | |
| Bartol et al. [1] | 25.4 |
| Ours ($H = 20$, $K = 10$) | **11.2** |

## 4.2 Implementation Details

To ensure fair comparison with previous studies, we use CPN [4] to extract 2D poses for the Human3.6M dataset and a more powerful 2D pose detector [49] for the CMU Panoptic dataset. The proposed method is implemented in PyTorch [30] using the AdamW optimizer [26], with momentum parameters $\beta_1 = 0.9$ and $\beta_2 = 0.999$, and weight decay of 0.1. Our model is trained for 100 epochs with an initial learning rate of $6e^{-5}$, decayed by a factor of 0.993 after the 10th epoch. During training, the number of hypotheses and iterations are set to $H = 1$ and $K = 1$, respectively. During inference, $H = 20$ and $K = 10$. The maximum diffusion time step $T$ is 1000. To avoid spurious local minima, we adopt geometry-guided sampling

starting from the second diffusion step. The guidance strength $s$ in Eq.(11) can be set manually. In our paper, $s = 1.0$. All experiments were conducted on a single NVIDIA GeForce RTX 3090 GPU.

## 4.3 Comparison with State-of-the-art Methods

**Results on Human3.6M.** As shown in Table 1 (Top), we compare Masked Gifformer with the state-of-the-art deterministic multi-view 3D HPE methods on Human3.6M. Following [1], the table only displays methods trained and tested on Human3.6M without additional training data (thus excluding [17]). For fairness, we set $H = 1$ to generate a single prediction. Although Masked Gifformer is designed to generate multiple 3D pose hypotheses, it achieves 24.7mm under the single hypothesis scenario, outperforming most deterministic methods and reaching performance comparable to the state-of-the-art [28]. Additionally, we compare Masked Gifformer with multi-view probabilistic methods, as shown in Table 1 (Bottom). When the number of hypotheses $H$ increases to 20, our method reduces the MPJPE from 24.7mm to 17.4mm. This demonstrates that generating multiple plausible pose hypotheses can alleviate the uncertainty still present in multi-view scenarios, thereby achieving more accurate pose estimation. Our results outperform [1] ($H = 200$) by a large margin with 11.7mm, even when using fewer hypotheses. To further validate the generality of our proposed Geometry-guided Sampling (GS), we apply it to the single-view probabilistic method [33]. As shown in Table 1 (Middle), with the help of GS, MPJPE of [33] decreased by 1.7mm. This indicates that incorporating geometric information during the sampling process can further enhance the accuracy of generated hypotheses.

**Table 3: Generalization performance on Human3.6M for different camera numbers and spatial placements. Best in bold.**

| Methods | Training on 2-view | | | | Training on 4-view | | |
|---|---|---|---|---|---|---|---|
| | 2-view | 2-view (unseen) | 3-view | 4-view | 2-view | 3-view | 4-view |
| Jiang et al. [18] | 29.5 | - | 42.4 | 39.2 | 32.6 | 29.3 | 27.8 |
| Shuai et al. [34] | 35.1 | 46.1 | 41.5 | 39.0 | - | - | 27.5 |
| Ours | **21.0** | **27.3** | **28.4** | **26.8** | **20.0** | **18.9** | **17.4** |

**Table 4: Generalization performance from CMU Panoptic to Human3.6M. Best in bold.**

| Methods \ Training set | CMU1 | CMU2 | CMU3 | CMU4 |
|---|---|---|---|---|
| Iskakov et al. [17] | - | - | 34.0 | - |
| Bartol et al. [1] | 33.5 | 33.4 | 31.0 | 32.5 |
| Jiang et al. [18] | 30.8 | 29.1 | 29.2 | 31.1 |
| Ours | **22.2** | **22.9** | **23.6** | **25.7** |

**Results on CMU Panoptic.** Table 2 compares our method with others on CMU Panoptic. Under a single hypothesis scenario ($H = 1$), Masked Gifformer surpasses the majority of deterministic methods. When $H$ increases to 20, MPJPE of our method drops to 11.2mm, also outperforming [1]. This is attributed to the incomplete views across most CMU Panoptic cameras, resulting in significant occlusions and missing body parts. Our method can provide more feasible poses, thus being more robust to occluded scenarios.

## 4.4 Generalization Performance

In addition to strong multi-view pose fitting capability, another ability of Masked Gifformer is that it can generalize well to varying cameras numbers, spatial arrangements, and even different datasets, which has been a primary limitation of previous methods.

**Generalization performance on different camera numbers and spatial placements.** The generalization results of Masked Gifformer and competitive methods on Human3.6M are shown in Table 3. Masked Gifformer is trained with 2 or 4 views and tested with varying numbers of views. From a horizontal analysis perspective, our method can successfully generalize to any number of cameras with minimal generalization loss ($< 7.4mm$). We observe that the more cameras used during training, the smaller the generalization gap. Specifically, the maximum generalization gap reduces to 2.6mm when trained with 4 views. Moreover, our method can generalize to different spatial arrangements (Table 3 (Left)). From a vertical analysis perspective, our method exhibits stronger generalization than [18] and [34] without sacrificing fitting abilities.

**Generalization performance on different datasets.** To test the cross-dataset generalization capability of Masked Gifformer, we follow [1, 18] to divide the CMU Panoptic dataset into four subsets, denoted as CMU1 ~ CMU4. Table 4 presents generalization results from four CMU Panoptic subsets to Human3.6M. From a horizontal comparison, our method successfully achieves cross-dataset generalization regardless of the training set. From a vertical comparison, our method demonstrates stronger generalization capabilities, surpassing [1, 17, 18]. For instance, when trained on CMU3 but tested on Human3.6M, MPJPE of [17], [1] and [18] are 34.0mm, 31.0mm, and 29.2mm, respectively. In contrast, our method achieves 23.6mm, indicating a notable improvement in cross-dataset generalization.

**Table 5: Ablation study on the components of our framework. We add each feature in turn and test on Human3.6M using different training datasets.**

| Fully random mask | | - | - | ✓ | - | ✓ |
|---|---|---|---|---|---|---|
| Geometry-guided sampling | | - | - | - | ✓ | ✓ |
| Multi-hypothesis | | - | ✓ | ✓ | ✓ | ✓ |
| **Train** | **Test** | | | | | |
| Human3.6M | Human3.6M | 23.2 | 24.7 | 22.7 | 18.2 | 17.4 |
| CMU3 | Human3.6M | 37.9 | 36.9 | 29.1 | 30.2 | 23.6 |

## 4.5 Ablation Study

**Effectiveness of each component.** We evaluate the effectiveness of our method by incrementally adding its components. Table 5 presents evaluation results on Human3.6M, including self-fitting results and cross-dataset generalization results. We begin with a naive combination of DMs and Transformers (1st column), and then utilize the inherent probabilistic nature of DMs to generate multiple pose hypotheses (2nd column). When $H$ increases to 20, MPJPE on Human3.6M decrease by 1.5mm and 1mm when trained on Human3.6M and CMU3 respectively. This result indicates that probabilistic methods can mitigate uncertainties in 3D HPE by generating multiple hypotheses. On one hand, we introduce the fully random mask mechanism into Transformer (3rd column). Under the multi-hypothesis setting, our method significantly improves the generalization performance ($36.9mm \rightarrow 29.1mm$) without sacrificing the fitting ability. This result indicates that fully random mask mechanism can substantially enhance the generalization across different datasets. On the other hand, we incorporate geometry-guided sampling into the diffusion model (4th column). MPJPE decreases to 18.2mm when multiple hypothesis are used. This result indicates that introducing geometric information into the sampling process can guide it towards minimizing reprojection errors, thereby achieving higher precision. Finally, combining the fully random mask and geometry-guided sampling (5th column) balances accuracy and generalization, achieving 17.4mm for the base dataset and 23.6mm for the cross dataset. These results also show the synergistic effect of the fully random mask and geometry-guided sampling.

**Analysis on fully random mask mechanism.** Fully random mask is designed to enhance the generalization capability. To validate its effectiveness, we train all models on Human3.6M with 4 views and set mask rates $M$ to 0, 0.2, 0.5, 0.8 and 1.0. As $M$ increases, more features from different views are discarded during training. $M = 0$ indicates that features from all views are involved in the fusion process. During inference, we evaluate Masked Gifformer under vaying mask rates by using test samples with different numbers of views, as shown in Table 6 (3rd column). Masked Gifformer generalizes well to any number of views when $M \in (0, 1)$, achieving

**Table 6: Generalization capability of different attention masks with varying mask rate $M$. Geometry-guided sampling is removed for a fair comparison. In the third column, all the models are trained on Human3.6M with 4 views and tested with 2,3 and 4 views. In the last column, all the models are trained on CMU3 and tested on Human3.6M. Best in bold.**

| Methods | $M$ | Number of Views $N$ | | | CMU3 → H36M |
|---|---|---|---|---|---|
| | | 2 | 3 | 4 | |
| w/o mask | 0 | 141.2 | 84.7 | 23.2 | 36.9 |
| Shuai et al. [34] | 0.2 | 27.5 | 25.2 | 22.9 | 38.4 |
| Ours | | 27.4 | 24.9 | 22.8 | 35.7 |
| Shuai et al. [34] | 0.5 | 26.4 | 24.9 | **22.7** | 37.2 |
| Ours | | **26.0** | **24.6** | **22.7** | **29.1** |
| Shuai et al. [34] | 0.8 | 28.5 | 26.7 | 24.4 | 33.0 |
| Ours | | 27.2 | 25.7 | 23.9 | 34.0 |
| Shuai et al. [34] | 1.0 | 65.7 | 64.5 | 65.4 | 29.6 |
| Ours | | 123.6 | 100.1 | 22.9 | 34.6 |

**Table 7: Ablation study on sampling process.**

| | Unconditional | Gaussian | Uniform | Ours |
|---|---|---|---|---|
| MPJPE (mm) ↓ | 22.7 | 20.2 | 19.1 | 17.4 |

optimal performance with $M = 0.5$. Furthermore, we compare our fully random mask with the random mask in [34]. On one hand, our method exhibits a lower generalization gap when adapting to different numbers of views. When $M = 0.5$, although both methods achieve $22.7mm$ when testing with 4 views, MPJPE of our method for 2 and 3 views is only $26.0mm$ and $24.6mm$, respectively, which is $0.4mm$ and $0.3mm$ lower than [34]. The greater advantage of the fully random mask is manifested in its cross-dataset generalization capability. We train all models on CMU3 and test them on Human3.6M. Trained on CMU3 and tested on Human3.6M, our method improves from $36.9mm$ to $29.1mm$, surpassing [34] by $8.1mm$ (Table 6, last column). Therefore, with the proposed fully random mask mechanism, our model demonstrates strong robustness to varying camera numbers, spatial arrangements and cross-datasets.

**Analysis on geometry-guided sampling process.** We further analyze the proposed geometry-guided sampling by exploring alternative methods such as uniform sampling and gaussian sampling, as shown in Table 7. Compared to unconditional sampling, both gaussian-guided sampling and uniform-guided sampling reduce the MPJPE to some extent, but not as effectively as our proposed geometry-guided sampling ($22.7mm → 17.4mm$).

**Analysis on JTB and MTB.** JTB learns the spatial structure of joints while MTB captures multi-view information. In JTB, each joint is treated as a token, outputting $X \in \mathbb{R}^{J \times N \times C}$ ($C$ is the number of hidden dimensions). Then, MTB integrates view information into the learned representations. Unlike previous methods, we separate different joints along the view dimension, treating the views of each joint as a single token $p \in \mathbb{R}^{1 \times N \times C}$. We model different joints in parallel before concatenating them. This can reduce dimensions from $J \times C$ to $C$, enabling to handle more views. We conduct an ablation study in Table 8. The difference between the last two columns is that our method alternately learns JTB and MTB. These results clearly demonstrate the effectiveness of JTB and MTB.

**Table 8: Ablation study on JTB and MTB.**

| | JTB only | MTB only | JTB+MTB | Ours |
|---|---|---|---|---|
| MPJPE (mm) ↓ | 38.0 | 181.4 | 19.0 | 17.4 |

Shan et al. [33]  [33]+GS (Ours)  Bartol et al. [1]  Ours

**(a) Qualitative results of related methods on Human3.6M**

2 views       3 views       4 views       Ground-truth

**(b) Qualitative results of various views on Human3.6M**

CMU1       CMU2       CMU3       CMU4

**(c) Qualitative results on CMU Panoptic**

**Figure 5: Qualitative results. Red line: predicted 3D pose. Blue line: ground truth 3D poses. Note that (b) shows the results trained on 4 views and then generalized to other views.**

## 4.6 Qualitative Evaluation

Qualitative results are shown in Figure 5. Consistent with quantitative results, Figure 5(a) demonstrates superior qualitative performance of our model compared to [33] and [1] on Human3.6M. Figure 5(b) shows evaluation results when generalizing to different numbers of views on Human3.6M. Figure 5(c) presents visualization results when our model is trained on different subsets of CMU Panoptic. These results highlight the effectiveness of our model across various datasets and its excellent robustness to varying camera numbers and spatial arrangements.

## 5 Conclusion

This paper presents Masked Gifformer, a novel diffusion model based on transformers for robust multi-view 3D HPE. During training, a transformer-based denoiser is supervised to fit 3D pose distribution by fusing joint and view information hierarchically. With the proposed fully random mask mechanism, Masked Gifformer is robust to varying camera numbers, spatial arrangements and datasets. During the inference, by modeling the conditional guidance distribution, multiple 3D pose hypotheses are sampled towards minimizing reprojection errors. Experimental results on two benchmarks show that our method effectively achieves a trade-off between accuracy and generalization.

# Acknowledgments

This work was partly supported by the National Science Foundation of China (grants no.U23B2030) and the Special Foundations for the Development of Strategic Emerging Industries of Shenzhen (Nos.JSGG20211108092812020 & KJZD20231023094700001).

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
