# OpenReview forum: "Geometry-Guided Diffusion Model with Masked Transformer for Robust Multi-View 3D Human Pose Estimation"
_acmmm.org/ACMMM/2024/Conference — MM2024 Poster_

### Official Review · Reviewer_TsXj · 2024-05-16

**Rating:** 3
**Confidence:** 3

**Summary:**

The paper proposes a Geometry-guided Diffusion Model with Masked Transformer (Masked Gifformer) for robust multi-view 3D HPE.

**Strengths:**

To address the long-standing problem of poor generalization, the paper introduces a fully random mask mechanism without any additional learnable modules or parameters; Authors incorporate geometric guidance into the diffusion model to enhance the accuracy of
the model.  The experiments on two popular 3D HPE benchmarks demonstrate the effectiveness of the proposed model.

**Limitations:**

(1) There are still some abbreviations and symbols are not clearly indicated and are not explained in detail, that make confusion, for example：a, line506-507：SO(3)，b. \lambda in equ(9), c. line 140 and equ (11): ∇_X, etc.
(2)Table 5 needs to be remade, as it is difficult to understand.
(3)There are some obvious formatting errors in the text. e,g., Highlight the vocabulary errors in bold, etc.
(4) Figure 3 should declare the meaning of X, Y axis and the meaning of different color in mask.
(5) Should provide the code, which helps to read the detail of the model.
(6) lack of novelty, the paper just combines the diffusion model with the transformer model, but the shortage of these model need more computing resource, AND the structure of JTB, MTB is just traditional transformer structure, Can these joint position information be effectively captured.

**Suitability:**

2

---

### Official Review · Reviewer_E5Lz · 2024-05-22

**Rating:** 4
**Confidence:** 3

**Summary:**

This paper introduces Masked Gifformer, a geometry-guided diffusion model designed for robust multiview 3D human pose estimation. The model comprises a hierarchical multiview transformer-based denoiser and incorporates a random mask mechanism. Furthermore, it integrates geometry guidance to enhance accuracy by modeling a conditional guidance distribution. Experimental results demonstrate its effectiveness across various datasets, outperforming other probabilistic methods. It achieves competitive results with state-of-the-art methods while maintaining good robustness.

**Strengths:**

1. The idea presented in this paper is novel. It introduces the concept of building distributions from geometry to serve as guidance for diffusion models, thereby enhancing their accuracy.
2. The experimental results are compelling. They demonstrate a significant improvement in the accuracy of generated methods, particularly notable given the challenge of stability during generation. Moreover, the model exhibits robustness, a critical factor in the field of pose estimation.
3. The paper is well-written. Its motivation is clear, and the methods are presented in an easy-to-follow manner. Overall, it is highly readable.

**Limitations:**

1. Expand Comparison on Robustness: In Table 3, only two multi-view methods are provided. Comparing with additional methods would significantly enhance the value of the paper, as robustness is a critical aspect of Human Pose Estimation (HPE).
2. Detailed Explanation of Experimental Results: Provide a thorough explanation of the experimental results. For example, clarify why the best results are achieved when the mask rate M is 0.5.

**Suitability:**

3

---

### Official Review · Reviewer_e8FK · 2024-05-23

**Rating:** 5
**Confidence:** 3

**Summary:**

This paper introduces Masked Gifformer, a Geometry-guided Diffusion Model with a Masked Transformer, for improved multi-view 3D HPE. The model uses a hierarchical multi-view transformer-based denoiser within a diffusion framework to integrate joint and view information for accurate 3D pose estimation. To enhance generalization, it employs a fully random mask mechanism without extra learnable modules. Geometric guidance is incorporated to optimize accuracy by minimizing reprojection errors.

**Strengths:**

1. The paper demonstrates the strong performance of diffusion-based models for 3D HPE, marking the early study to evaluate and report the effectiveness of diffusion models in this domain, highlighting their potential for accurately capturing the complex 3D pose distributions.

2. The introduction of a fully random mask mechanism within the multi-view transformer block significantly enhances the model's robustness to variations in camera numbers, spatial arrangements, and different datasets. This mechanism achieves this without the need for any additional learnable modules or parameters, maintaining model simplicity and efficiency.

3. Integrating geometric information into the sampling process of diffusion models improves the accuracy of the 3D pose estimation. This is achieved by optimizing the diffusion process to minimize reprojection errors, effectively guiding the model to generate more precise and geometrically consistent poses.

**Limitations:**

1. More analysis is needed to clearly demonstrate the advantages of probabilistic methods over deterministic ones. This could include a detailed comparison of performance metrics, robustness to noise, and ability to capture uncertainty in pose estimation.

2. Although the performance improvements in multi-view settings are not very strong, a deeper exploration into other metrics where deterministic methods may underperform, such as calibration accuracy and handling of occlusions, could provide a more comprehensive evaluation of the model's benefits.

3. The paper would benefit from further analysis of the Geometry-guided sampling process. Exploring alternative approaches, such as uniform sampling or implementing a learnable or tunable sampling prior, could offer insights into optimizing the sampling process and potentially enhance the model's accuracy and flexibility.

**Suitability:**

3

---

### Official Review · Reviewer_ZkuQ · 2024-05-26

**Rating:** 3
**Confidence:** 2

**Summary:**

The authors aim to improve the accuracy and generalization of 3D human pose estimation from multi-view images. The authors introduce a Geometry-guided Diffusion Model with a Masked Transformer (Masked Gifformer) that integrates joint and view information using a hierarchical multi-view transformer-based denoiser. To enhance generalization, the model incorporates a fully random mask mechanism, which avoids overfitting to specific camera numbers and positions. Additionally, geometric guidance is used in the sampling process to minimize reprojection errors, further improving accuracy. Extensive experiments verify the effectiveness of the proposed method.

**Strengths:**

- In general, this paper is written well. The method is well explained with sufficient illustrative diagrams (such as Fig.1,2).
- The ablation studies show the effectiveness of the proposed method.

**Limitations:**

- Incremental Contribution: This paper heavily relies on previous diffusion model-based 3D HPE methods that use 2D key points, such as [1]. The proposed geometric-guided sampling and random mask mechanisms are minor improvements and less impressive, falling short of the high standards required for acceptance in ACM-MM.

- The paper seems to lack a detailed demonstration of the Joint Transformer Block (JTB) and the Multi-view Transformer Block (MTB) in the "Denoiser" module. These components are not thoroughly discussed in the text or included in the ablation studies, leaving uncertainty about their contributions. This requires clarification.

- As shown in Table 6, the generalization capability gained by the proposed mask mechanism is not significantly better compared to the method in [34]. The improvement is marginal, which raises concerns about the effectiveness of the proposed approach.

[1] Diff3DHPE: A Diffusion Model for 3D Human Pose Estimation. CVPR'2023

**Suitability:**

3

---

### Meta-Review · Area_Chair_uaRB · 2024-07-05

**Recommendation:** Accept (Poster)
**Confidence:** 4

**Metareview:**

After considering all reviews, the rebuttal, and the subsequent discussion, the consensus is to accept the paper.